# TADfit is a multivariate linear regression model for profiling hierarchical chromatin domains on replicate Hi-C data

Erhu Liu [1], Hongqiang Lyu [1,2✉], Qinke Peng[1], Yuan Liu[1], Tian Wang[3] & Jiuqiang Han[1,2]

Topologically associating domains (TADs) are fundamental building blocks of three dimensional genome, and organized into complex hierarchies. Identifying hierarchical TADs on Hi-C data helps to understand the relationship between genome architectures and gene regulation. Herein we propose TADfit, a multivariate linear regression model for profiling hierarchical chromatin domains, which tries to fit the interaction frequencies in Hi-C contact matrix with and without replicates using all-possible hierarchical TADs, and the significant ones can be determined by the regression coefficients obtained with the help of an online learning solver called Follow-The-Regularized-Leader (FTRL). Beyond the existing methods, TADfit has an ability to handle multiple contact matrix replicates and find partially overlapping TADs on them, which helps to find the comprehensive underlying TADs across replicates from different experiments. The comparative results tell that TADfit has better accuracy and reproducibility, and the hierarchical TADs called by it exhibit a reasonable biological relevance.

[1] School of Automation Science and Engineering, Faculty of Electronic and Information Engineering, Xi'an Jiaotong University, Shaanxi 710049, China.
[2] Guangdong Artificial Intelligence and Digital Economy Laboratory, Guangdong 510335, China. [3] Institute of Artificial Intelligence, Beihang University, Beijing 100191, China. ✉email: hongqianglv@mail.xjtu.edu.cn

Previous studies suggest that the three-dimensional architecture of eukaryotic genome is non-random and organized into complex stratums[1–4]. Knowledge about genome architectures is helpful for understanding multiple cellular processes, such as epigenetic organization[5], gene regulation[6–8], and DNA replication timing[9]. As a derivation of chromosome conformation capture technology, Hi-C has been widely used to investigate the spatial organization of genome for its ability in profiling chromatin interactions on a genome-wide scale. It produces up to billions of paired-end reads, which can be binned into a contact matrix[10], and the element in this contact matrix reflects the interaction frequency (IF) between the corresponding pair of genomic loci. Hi-C technology has brought a deeper insight into the chromatin organization at multiple levels, including A/B compartment[10,11], topologically associating domain (TADs)[3,4], and chromatin loops[7,12]. Among them, TADs are structural blocks composed of genomic regions that show a high degree of self-interacting, and play an important role in guiding and constraining long-range regulation of gene expression[13–15]. Getting off the ground, TADs were regarded as disjoint functional blocks of genome, which makes the concerns focused on their boundaries. It was observed that TAD boundaries are enriched for insulator binding protein CTCF, housekeeping genes, specific cohesion complexes, and histone marks[3,16], and the disruption of these boundaries may result in misregulation and even diseases[17,18]. Further studies showed that TADs exhibit structural heterogeneity and functional diversity, which cannot be explained with traditional disjoint organization of TADs[19–21]. Thus, the internal substructures, known as sub-TADs nested within meta-TADs, were subsequently investigated[21–23]. It was found that these hierarchical sub-TADs have different transcriptional activities, they are segregated into compartment A or B, and exhibit distinctive epigenetic features, such as histone marks and gene expression level, while the larger meta-TADs appear to be more stable[24,25]. Currently, some partially overlapping TADs have also been reported[26–28]. They are considered to be associated with weak border or transition zone of two adjacent TADs, and may play an important role in gene regulation[29].

A variety of computational methods have been proposed to identify TADs on Hi-C contact matrix. According to whether the hierarchical structures of TADs are considered, they can be roughly divided into two categories. One category regards TADs as insulated regions without mutual containing or overlapping, such as Directionality Index (DI)[3], HiCSeg[30], Insulation Score (IS)[31], TopDom[32], and ClusterTAD[33]. Among them, DI, IS, and TopDom are all linear score approaches, the difference is that DI scores each bin with a defined directionality index and infers TAD boundaries with the help of a hidden Markov model, while IS and TopDom determine boundaries by locating a local minimum insulation bin via a customized insulation square and diamond-shaped area, respectively. HiCSeg and ClusterTAD are separately statistical and clustering approaches, the former calculates an optimal segmentation by solving a maximum likelihood estimation problem with dynamic programming, and the later detects TAD boundaries using an unsupervised clustering algorithm[34]. The other category takes into account sub-TADs nested within meta-TADs, such as TADtree[22], GMAP[23], CaTCH[35], 3DNetMod[28], OnTAD[25], SpectralTAD[36], and TADpole[37]. Among them, TADtree and GMAP both rely on statistical models of the interaction distributions, to the best of our knowledge, the former is the first publicly available approach to detect nested TADs by optimizing an objective function that describes the hierarchy of TADs, while the later achieves the same purpose by combining Gaussian mixture model with proportion test. CaTCH and OnTAD are considered to be hybrid linear score approaches which determine nested TADs from candidate ones with the help of customized score functions. 3DNetMod is a graph-theory-based approach which can handle nested and partially overlapping TADs by optimizing network modularity. And SpectralTAD and TADpole are clustering approaches, the former determines nested TADs via spectral clustering algorithm, while the later via constrained hierarchical clustering. The two categories of methods are devoted to the identification of disjoint and nested TADs, respectively. Unfortunately, the existing methods are designed to accept individual Hi-C sample, none of them is capable of handling multiple replicates. Besides, these methods assume that TADs are disjoint or nested without considering more complex structures, except for 3DNetMod which can find out partially overlapping ones by optimizing network modularity based on graph theory[28]. With the continuous accumulation of Hi-C datasets and introduction of partially overlapping TADs, it may help to get a deeper understanding of genome architectures at TAD level to outline a more comprehensive profile of hierarchical chromatin domains on the basis of replicate Hi-C data.

In this paper, we present TADfit, a multivariate linear regression model for profiling hierarchical chromatin domains. It tries to fit the IFs in Hi-C contact matrix with and without replicates using all-possible hierarchical TADs, and the regression coefficients, which can be used to determine the significance of these TADs, are obtained by an online learning solver called Follow-The-Regularized-Leader (FTRL)[38,39]. Beyond the existing methods for TAD identification, TADfit addresses the following two issues, one is an attempt to accept replicate Hi-C data, which is conducive to find the underlying TADs across different experiments. The other one is that it runs without any assumption that TADs are disjointed or nested, so that more comprehensive structures at TAD level can be revealed, such as partially overlapping ones. Using both simulated and experimental Hi-C data, the two issues were demonstrated by a comparative analysis with the other five state-of-the-art methods, including TADtree, 3DNetMod, OnTAD, SpectralTAD, and TADpole, and the results tell that TADfit has better accuracy and reproducibility in almost all cases. Besides, the hierarchical TADs called by it present a reasonable biological relevance in terms of histone marks, architectural proteins, regulatory elements, gene expression level and A/B compartment, which suggests that TADfit profiles a biologically relevant hierarchy of TADs.

## Results

**Overview of TADfit**. TADfit is designed to fit the IFs of Hi-C contact matrix replicates with all-possible hierarchical TADs on them using a multivariate linear regression model. It takes a group of Hi-C contact matrix replicates or an individual contact matrix as input, and outputs the regression coefficient for each hierarchical TAD. TADfit consists of three major steps (Fig. 1). The first step, preparation of candidate hierarchical TADs. A pseudo contact matrix is generated by geometric mean per matrix element across replicates, The TAD boundaries called by TopDom on diagonal of the pseudo contact matrix are then assembled in all-possible pairs, and the chromatin domain between each pair of boundaries is regarded as a candidate TAD. In this way, the prepared candidate TADs can cover all the possible hierarchical structures at TAD level, including disjoint, nested, and partially overlapping ones. The second step, modeling the relationship between IFs and candidate hierarchical TADs. A multivariate linear regression model is proposed to describe the relationship between the IFs of contact matrix replicates and the candidate hierarchical TADs, on the hypothesis that each IF in contact matrix reflects the cumulative effect of hierarchical TADs in which it is fallen. And the unknown regression coefficients which reflect the weights of candidate TADs can be estimated by solving an

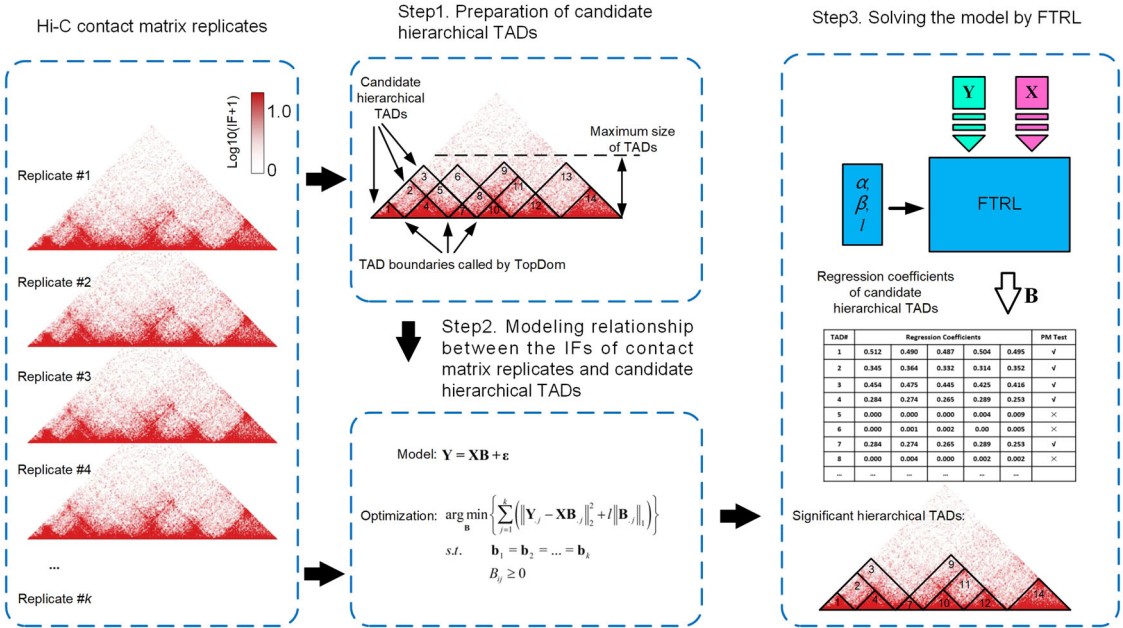

**Fig. 1 Overview of TADfit.** TADfit consists of three major steps. Step 1, a set of TAD boundaries on the diagonal of a pseudo contact matrix derived from input replicates is called by TopDom, and then assembled in all-possible pairs to prepare the candidate hierarchical TADs. For example, a total of 21 candidate hierarchical TADs can be obtained by assembling 7 TAD boundaries, and the number of these TADs can be optionally limited to 14 according to the threshold of TAD size given by user. Step 2, a multivariate linear regression model is proposed to fit the IFs of Hi-C contact matrix replicates with the candidate hierarchical TADs, on the hypothesis that each IF in contact matrix reflects the cumulative effect of hierarchical TADs in which it is fallen. Step 3, the model is solved with the help of FTRL. The regression coefficients for these candidate hierarchical TADs are output, and a right-tailed permutation test is used to determine the significant ones, as shown above, 10 significant ones are screened out from 14 candidate TADs.

optimization problem characterized by a minimum objective function. The third step, solving the model by FTRL. An online machine learning algorithm called FTRL is chosen to solve the optimization problem, so that the regression coefficients for these candidate hierarchical TADs are obtained, and the significant ones can be screened out by a right-tailed permutation test. Beyond the existing methods, TADfit has an ability to handle multiple contact matrix replicates and find partially overlapping TADs on them. The details can be found in the "Methods" section.

**Results on simulated Hi-C data.** To investigate the performance of TADfit in identifying hierarchical TADs on simulated Hi-C data, the simulated contact matrix replicates, with and without partially overlapping TADs at five different noise levels, were fed into TADfit and the other five methods, including TADtree, 3DNetMod, OnTAD, SpectralTAD, and TADpole. The called TADs were compared with ground-truth TADs using Jaccard index and F1 score to examine the accuracy of these methods in identifying hierarchical TADs in different cases. We first ran the proposed TADfit on simulated Hi-C data in two different ways. One way is to feed a group of contact matrix replicates into TADfit to get a set of called TADs across replicates, the other is to feed one individual contact matrix each time to get a respective set of TADs for each replicate. It is obvious that the former has a higher Jaccard index and F1 score than the later (Supplementary Tables 1, 2). That tells the advantages of taking multiple replicates into account in the design of our model. Moreover, even in the case of processing individual contact matrix one by one, TADfit can produce much higher Jaccard index and F1 score than the other five methods, regardless of the noise level and whether partially overlapping TADs are considered (Fig. 2a, b). In addition, to conduct an intuitive comparison, heatmaps with the called hierarchical TADs on them at five noise levels were presented. As we can see, for simulated Hi-C data with partially

overlapping hierarchical TADs, no method can decompose TADs when they are partially overlapped, except for TADfit and 3DNetMod (Fig. 2c and Supplementary Figs. 1–5), where the former outlines almost all the hierarchical TADs, including partially overlapping ones, while the latter tends to give out some small sub-TADs which are not consistent with the ground-truth. For simulated Hi-C data without partially overlapping TADs (Supplementary Figs. 6–10), all the above methods have the ability to outline disjoint and nested TADs, but their power is different. TADtree and OnTAD may overlook some unobvious sub-TADs, 3DNetMod and SpectralTAD, on the contrary, are inclined to produce some excessively small sub-TADs, and TADpole prefers to regard some large but visually unreliable regions as TADs, leaving TADfit to describe hierarchical structures better in most cases. Thus, TADfit outperforms the other five methods in identifying hierarchical TADs on simulated data.

**Regression analysis of the proposed model.** To gain insight beyond simulation, a regression analysis was conducted on experimental Hi-C data to investigate the ability of our proposed model in interpreting the true relationship between interaction frequencies and candidate hierarchical TADs, as well as the effectiveness of FTRL online regression solver. The analysis results on five contact matrix replicates for chromosome 1 of GM12878 at 25K resolution were shown in Fig. 3. Obviously, the value of R-squared ($R^2$) grows as the number of iterations increases, and while FTRL online learning is iterated 10 times, the average $R^2$ can reach up to $0.89 \pm 0.02$ (Fig. 3a). That is to say, the variance of dependent variable can be well explained by the independent variables in our regression model. Considering computational economy, two iterations with an average $R^2$ of $0.86 \pm 0.02$ is acceptable, so that all the results in this paper were produced using this configuration arbitrarily. Moreover, the scatter plot of fitted values versus original values were presented

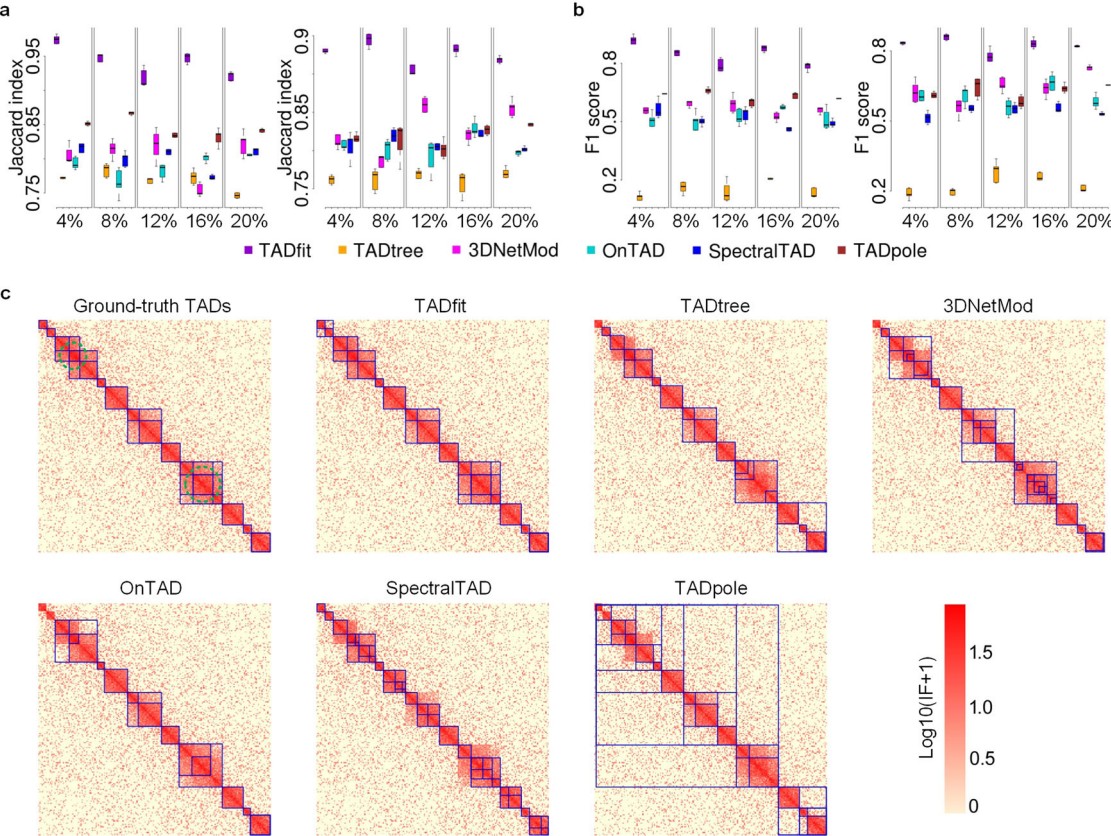

**Fig. 2 Performance comparison on simulated Hi-C data.** TADfit and the other five methods, including TADtree, 3DNetMod, OnTAD, SpectralTAD, and TADpole, were compared on simulated Hi-C data. Simulated contact matrices were fed into these methods individually, the called TADs on each contact matrix were compared with ground-truth TADs to obtain a set of Jaccard indexes and F1 scores. Boxplots of **a** Jaccard index and **b** F1 score on simulated contact matrices with (left) and without (right) partially overlapping TADs at five different noise levels (4%, 8%, 12%, 16%, 20%) were given. The center line of the box indicates the median, whereas the bottom and top of the box indicate the first and third quartiles, respectively, and whiskers are extended to the most extreme data point that is no more than 1.5 × interquartile range from the bottom and top of the box. Besides, **c** heatmaps of a simulated contact matrix (ChrS_MAT_noise0.20_POP0.15_rep1, bin 1–bin 200 out from a total of about 400 bins) and the TADs called by different methods on it were shown. These heatmaps were drawn on a log scale, the ground-truth TADs as well as the TADs called by TADfit and the other five methods were outlined with blue solid lines, and the partially overlapping TADs were marked with a green dotted circle on the first heatmap.

to visualize the statistical difference between them (Fig. 3b). In general, the scattered points are evenly distributed on both sides of the ideal baseline, and the loess-fitted curve runs close to this line, which is consistent with the above results scored by $R^2$. The homoscedasticity of residuals was also examined (Fig. 3c, d). It seems that the residuals are independent of fitted values, there should be no discernible pattern, and the proposed model has captured the inherent relationship between residuals and fitted values to a considerable extent. In addition, considering heatmap is the primary means of overall graphical presentation of Hi-C data, herein an artificial contact matrix with the upper right triangle for original values and the lower left triangle for the corresponding fitted values was visualized in the form of a heatmap (Fig. 3e). As we can see, the IFs in the two symmetrical triangular regions follows the same decay pattern in signal as bin distance increases, and most importantly, these two regions exhibit a highly similar block structure with each other, especially near the diagonal, which reflects the goodness of fitting of our model and the effectiveness of FTRL solver. Some extra analysis results for different cell lines (GM12878, IMR90, and K562) can be found in Supplementary Figs. 11–13.

**Reproducibility of the called TADs.** Different from simulated Hi-C data, the ground-truth TADs on experimental Hi-C data are

unknown, and there is no gold standard to score the accuracy of identified TADs. To evaluate the performance of TADfit in profiling hierarchical TADs on experimental contact matrices, the reproducibility of the called TADs was examined in multiple ways. First, reproducibility across replicates. Since TADfit is a multi-replicate method, the hierarchical TADs called by it would be exactly the same across replicates, in other words, the Jaccard index of hierarchical TADs called by TADfit between replicates should always be one. Thus, there is no doubt that the index is absolutely higher than those of the other methods where a multi-replicate input is not allowed. To conduct a comparative analysis on an equal footing, herein contact matrix replicates were fed into TADfit individually, just like the other methods. Even so, it can be seen that the Jaccard index of hierarchical TADs called by TADfit is still higher than those of the other five methods in most cases (Fig. 3f and Supplementary Fig. 14). Second, reproducibility between different resolutions. Genome is organized into complex hierarchical structures at TAD level. Chromatin domains at higher resolution can be regarded as a subdivision of the domains at lower resolution, and chromatin domains at lower resolution should be detectable in the corresponding regions at higher resolution. That is to say, the hierarchical TADs detected at lower resolution are theoretically a subset of those detected at higher resolution. Based on this, a modified Jaccard index is proposed to assess how the hierarchical TADs identified on contact matrices

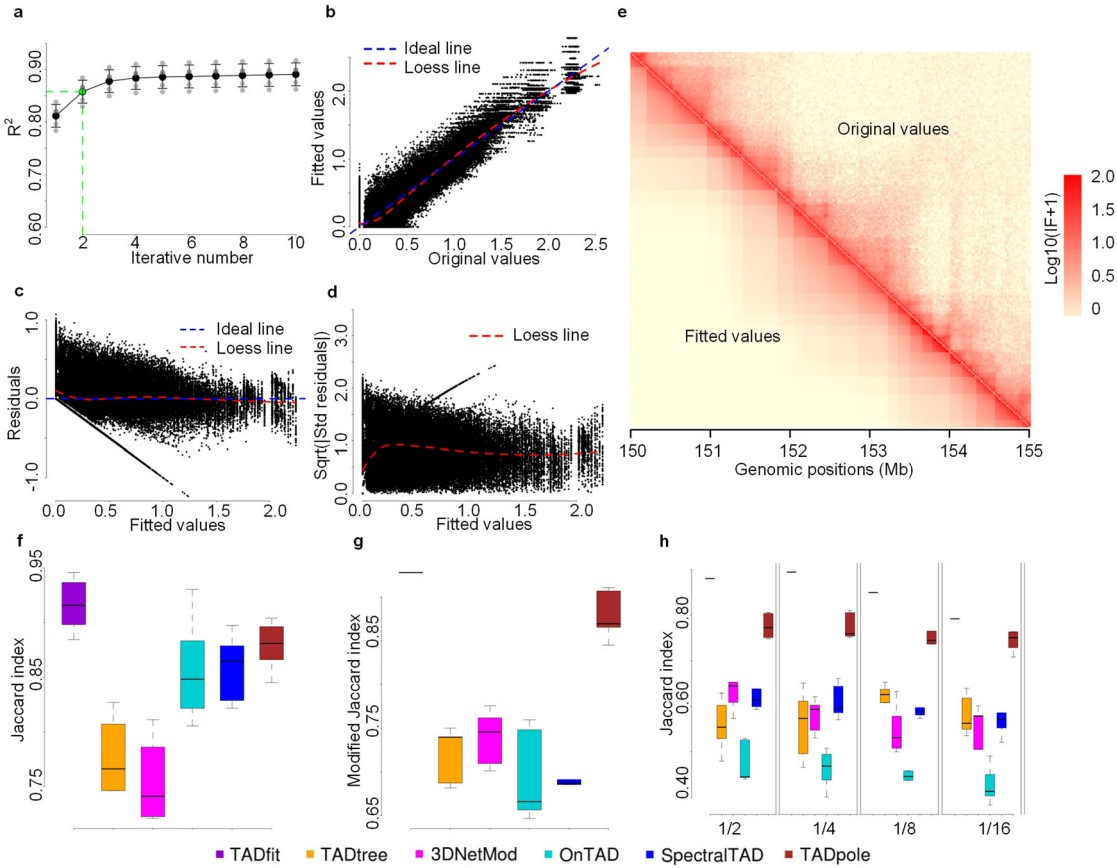

**Fig. 3 Performance evaluations on experimental Hi-C data.** In the absence of clear indication, the evaluations were carried out on contact matrix replicates (GSM1551550_HIC001–GSM1551554_HIC005) for chromosome 1 of GM12878 at 25K resolution. As one part, a regression analysis was conducted to investigate the goodness of fitting of our TADfit. **a** A curve of $R^2$ (mean ± SD) versus iterative number, as well as a total of three scatter plots, including **b** plot of fitted values versus original values, **c** plot of residuals versus fitted values and **d** scale location plot, were shown, and the loess-fitted curves were drawn with red dotted line. Besides, **e** the heatmap of an artificial contact matrix with the upper right triangle for original values (GSM1551550_HIC001, 150–155 Mb) and the lower left triangle for the corresponding fitted values was presented. As the other part, the reproducibility of hierarchical TADs called by TADfit and the other five methods in three different contexts were compared, including **f** reproducibility across replicates, **g** reproducibility between different resolutions (50K versus 25K), and **h** variation at four different sequencing depth levels (1/2, 1/4, 1/8, and 1/16). To conduct a comparative analysis on an equal footing in the first context, the contact matrix replicates were fed into TADfit individually, since TADfit is a multi-replicate method, the Jaccard index of hierarchical TADs called by it between replicates should always be one, which is always higher than those of the other methods where a multi-replicate input is not allowed. The reproducibility in these contexts was quantified using Jaccard index, except for the middle context where a modified Jaccard index is proposed to assess how the hierarchical TADs identified on contact matrices at lower resolutions are reproducible on corresponding contact matrices at higher resolutions. The center line of the box indicates the median, whereas the bottom and top of the box indicate the first and third quartiles, respectively, and whiskers are extended to the most extreme data point that is no more than 1.5 × interquartile range from the bottom and top of the box.

at lower resolutions are reproducible on corresponding contact matrices at higher resolutions (Fig. 3g and Supplementary Fig. 15). Compared with the other methods, TADfit and TADPole have higher modified Jaccard indexes. Finally, variation at different sequencing depths. Four artificial contact matrices were generated by down-sampling[40] each experimental contact matrix at four different sequencing depth levels (1/2, 1/4, 1/8, and 1/16), and the hierarchical TADs called by different methods on these artificial contact matrices were compared with those on corresponding original matrices using Jaccard index. In general, the index decreases with the level of sequencing depths comes down, and TADfit has higher Jaccard indexes in almost all cases (Fig. 3h and Supplementary Fig. 16).

**Biological relevance of TADs profiled by TADfit.** The biological features of hierarchical TADs profiled by TADfit were investigated with the help of histone marks, architectural proteins, gene

expression level, and A/B compartment. Different from the other methods to give out the hierarchical level of called TADs, TADfit can return a regression coefficient for each possible hierarchical TAD. Considering hierarchical structures, these coefficients were projected to corresponding chromatin regions according to the genomic coordinates of called TADs, and then accumulated, so that a comprehensive value (CV) was obtained for each basic TAD block, which is a chromatin region sandwiched between two adjacent TAD boundaries. As shown in Fig. 4, the CVs of these basic TAD blocks are different from each other, indicating the divergence between these relatively independent structures at TAD level. What is interesting is that the CVs of these basic TAD blocks have a considerable relevance with the degree of enrichment of active histone marks (H3K36me3, H3K4me3, and H3K27ac) and gene expression level, while the repressive histone marks (H3K27me3) show a weak opposite relevance. That is to say, in the same chromatin region, the greater the CVs, the more

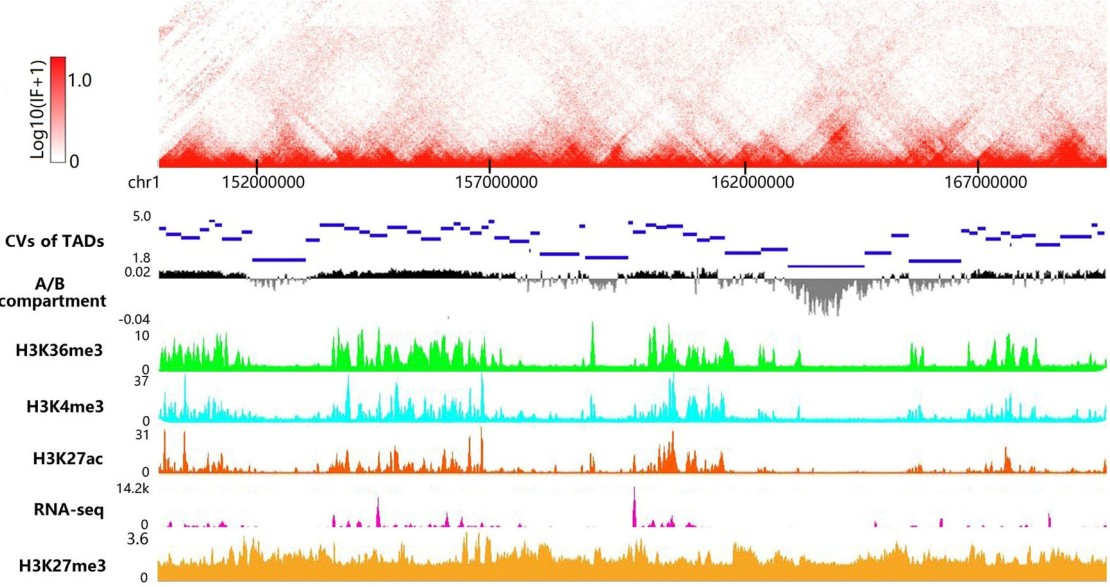

**Fig. 4 Biological features of hierarchical TADs called by TADfit.** The hierarchical TADs were called by TADfit on contact matrix replicates (GSM1551550_HIC001–GSM1551554_HIC005) for chromosome 1 of GM12878 at 25K resolution, the regression coefficients were projected to corresponding chromatin regions according to the genomic coordinates of called TADs, and then accumulated, so that the CVs for basic TAD blocks were obtained. For A/B compartment, the first eigenvector was calculated on replicate GSM1551550_HIC001 via an eigenvector command provided by juicer_tools[51]. The histone marks (H3K36me3, H3K4me3, H3K27ac, and H3K27me3) and RNA-seq data were captured online with the aid of New WashU Epigenome Browser[55] available at www.epigenomegateway.wustl.edu. This browser was also used to graphically present the above data in the form of heatmap (GSM551550_HIC001, 150–170 Mb) and tracks.

enriched the active histone marks, and the higher the level of gene expression. Besides, the regions with greater CVs are obviously more inclined to belong to active compartment A, and vice versa to repressive compartment B. In addition, the enrichment of architectural proteins (CTCF, Smc3, and Rad21) and histone mark (H3K4me3) within 50 Kb of TAD boundaries at different hierarchical levels was also examined across three cell lines (GM12878, IMR90, and K562) on a genome-wide scale (Supplementary Fig. 17). Herein the level of a TAD boundary is determined using the terminology given by An et al.[25], that is, a TAD boundary belonging to a single TAD is regarded as a first-level boundary, and the second-level and third-level boundaries correspond to the ones that are shared by two and three hierarchical TADs, respectively. It is shown that the number of ChIP-seq peaks at the second-level boundaries is significantly greater than that at the first-level boundaries (right-tailed paired $t$-test, $p$-value <5.04e−4), based on an assumption that the enrichment differences between two different levels are normally distributed irrespective of the type of ChIP bindings and cell lines, and the same trend can also be seen at the TAD boundaries between the higher levels and second-level (right-tailed paired $t$-test, $p$-value <6.22e−5), thus, the enrichment of these ChIP bindings at TAD boundaries is significantly enhanced as the level of boundaries increases, which is in line with the observations of other studies[21,36,41] where the TAD boundaries with higher levels are believed to be more biologically meaningful. These results tell that the regression coefficients and hierarchical level of TADs called by TADfit present a reasonable biological relevance to some extent.

## Discussion

To illustrate the ability of TADfit in handling partially overlapping TADs, the hierarchical TADs called by TADfit, OnTAD, and TADpole on a contact matrix were compared (Fig. 5a and

Supplementary Fig. 18). It is worth noting that a TAD marked with a green dotted circle on heatmap of the contact matrix can be considered as a sub-TAD nested within both an upstream meta-TAD and a downstream meta-TAD. That is to say, the TAD is located in the common overlapping area of two adjacent TADs at a higher hierarchical level, which seems to be a partially overlapping TAD. To this complex hierarchical structure, different methods have different responses. It can be seen that since OnTAD and TADpole are designed without considering partially overlapping structure, the former takes the TAD for a sub-TAD completely nested within the downstream meta-TAD, leaving the upstream meta-TAD unrevealed, while the latter is just the opposite, which takes the TAD for a sub-TAD completely nested within the upstream meta-TAD, leaving the downstream meta-TAD unrevealed. The inconsistent results indicate the limitations of such methods, and support the rationality of the existence of partially overlapping TADs to some extent. Unlike them, TADfit tries to fit the interaction frequencies using all-possible TADs without assuming that these TADs are disjointed or nested, so that both upstream and downstream meta-TADs can be identified even if they are partially overlapped. For the partially overlapping TADs called by TADfit, the reproducibility of them across contact matrix replicates was examined (Fig. 5b). The median values of Jaccard indexes are always higher than 0.60 in all cases, and the highest value can reach up to 0.87, even though lower than that of the other TADs. One more step forward, herein we tried to scan the whole genome of GM12878 at 25K resolution using our TADfit in the way of multi-replicate input, the numbers of both partially overlapping TADs and the other TADs for every chromosome are given (Supplementary Fig. 19). It is found that about 10.95% of the called TADs are partially overlapping ones. Thus, partially overlapping structure at TAD level may be considerable in spatial organization of human genome. That is basically in line with our expectations derived from other Hi-C experiments. In detail, super-resolution microscopy

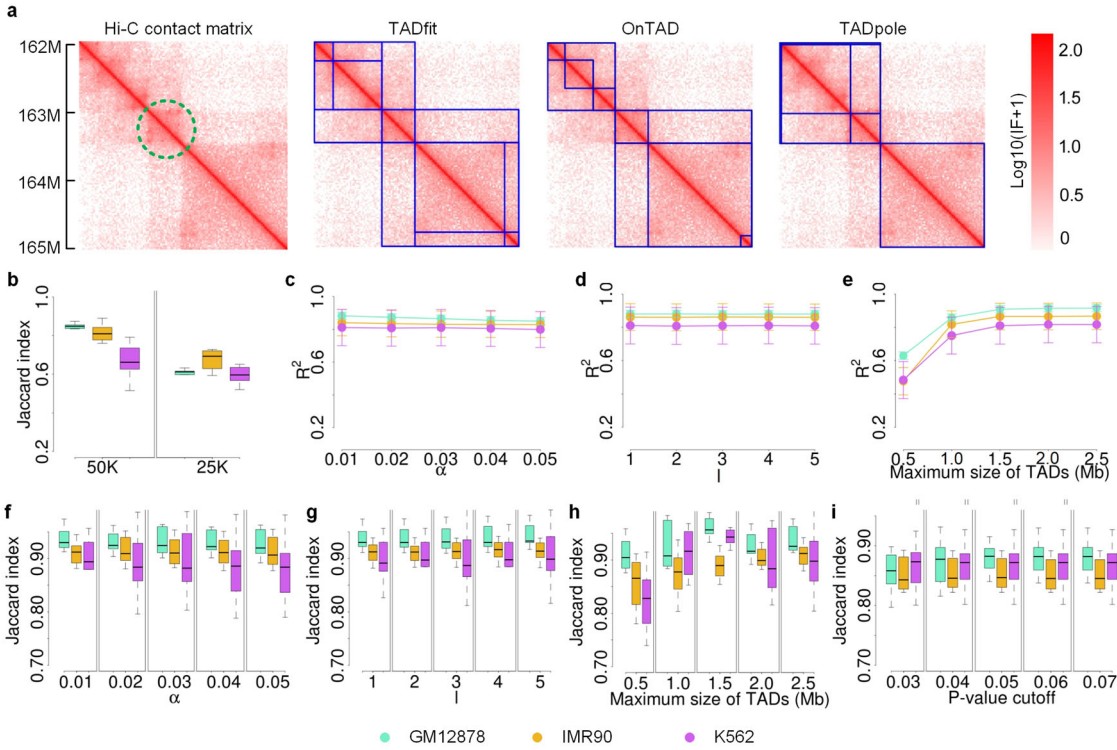

**Fig. 5 Illustration of TADfit in handling partially overlapping TADs and the robustness to hyperparameters.** To analyze the ability of TADfit in handling partially overlapping TADs, the hierarchical TADs were called by TADfit on contact matrix replicates (GSM1551550_HIC001–GSM1551554_HIC005) for chromosome 1 of GM12878 at 25K resolution, and one of the replicates (GSM1551550_HIC001) was fed into OnTAD and TADpole, since they cannot accept multiple replicates as input. **a** Heatmaps of contact matrix (GSM1551550_HIC001, 162 Mb–165 Mb) and the hierarchical TADs called by the three methods on it were drawn. There are four heatmaps shown above, on the first of which a partially overlapping TAD was marked with a green dotted circle, and on the last three of which the called TADs were outlined with blue solid line. Besides, to investigate the reproducibility of partially overlapping TADs called by TADfit, the contact matrix replicates for chromosome 1 of three cell lines (GM12878, IMR90, and K562) at resolutions of 50K and 25K were fed into TADfit individually, and **b** the Jaccard index of partially overlapping TADs across replicates were shown. In addition, to analyze the affection of the tuning of two hyperparameters ($\alpha$ for per-coordinate learning rate and $l$ for the strength of $L_1$ regularization) on the iterative optimization, as well as the tuning of two parameters (the maximum size of TADs and $p$-value cutoff of permutation test) on the identification of hierarchical TADs, the contact matrix replicates for chromosome 1 of three cell lines (GM12878, IMR90, and K562) at 25K resolution were separately fed into TADfit with each cell line as a group. **c–e** The curves of $R^2$ (mean ± SD) with different $\alpha$, $l$ and maximum sizes of TADs, as well as **f–i** the Jaccard index of called TADs across replicates with different $\alpha$, $l$, maximum sizes of TADs and $p$-value cutoffs were given, respectively. The center line of the box indicates the median, whereas the bottom and top of the box indicate the first and third quartiles, respectively, and whiskers are extended to the most extreme data point that is no more than 1.5 × interquartile range from the bottom and top of the box.

image of chromatin with nanometer-scale precision shows that TAD structures in a bulk Hi-C contact matrix can be considered as the ensemble average of TAD-like domains in thousands of single cells[42], and the boundaries of these TAD-like domains in single cells present a high degree of cell-to-cell variations[42–44], whereas it is suggested that the hierarchical TADs in bulk Hi-C contact matrix are unlikely to be perfectly nested, but also partially overlapped. This may be one of the motivations that partially overlapping structure has been considered now or claims to be in the further by a few of the latest computational methods for TAD identification[25,28,45]. In addition, the biological features between partially overlapping regions and the other regions were compared in terms of histone marks (H3K36me3, H3K4me3, H3K27ac, and H3K27me3) and active genes (FPKM >5) (Supplementary Fig. 20). Generally, an enrichment difference can be seen between the two types of regions by examining all the results throughout different cell lines (GM12878 and K562), that is, the average signal of active histone marks (H3K36me3, H3K4me3, and H3K27ac) and the density of active genes in partially overlapping regions trend to be higher than that of the other regions, while the repressive histone mark (H3K27me3) does the opposite. That can be explained, since active epigenetic states and highly

expressed genes are reported to be more enriched in inner TADs than in outer TADs[21,36,41,46], and the partially overlapping regions are usually inner parts in a hierarchy of TADs. Furthermore, although the mechanisms underlying hierarchical TADs, especially partially overlapping ones, in gene expression regulation remain unclear[47,48], the promoters and enhancers, especially active promoters and strong enhancers, are found to be favored within partially overlapping regions (Supplementary Fig. 21). That suggests the high-intensity interaction between enhancers and promoters within these regions and support the enrichment difference in active genes.

There are two hyperparameters need to be tuned during the iterative optimization of FTRL, including $\alpha$ for per-coordinate learning rate and $l$ for the strength of $L_1$ regularization strength. The former determines the step size at each iteration while moving toward a minimum of loss function, and the latter controls the sparsity of regression coefficients. To examine how significantly the tunning of these two hyperparameters affect the iterative optimization, the curves of $R^2$ versus $\alpha$ and $l$ were plotted, respectively (Fig. 5c, d). As we can see, the values fluctuate slightly with the increase of $\alpha$ from 0.01 to 0.05 and $l$ from 1 to 5, and the average $R^2$ is always larger than 0.80. These suggest

that the online iterative optimization is not significantly affected when the two hyperparameters change in a reasonable range. Besides, the reproducibility of hierarchical TADs called by TADfit across replicates were checked when $\alpha$ and $l$ change (Fig. 5f, g). It can be seen that the median values of Jaccard indexes of hierarchical TADs between replicates are larger than 0.85 in all cases, indicating the robustness of called TADs to the two hyperparameters. In addition, another two parameters in TADfit, including the maximum size of TADs and $p$-value cutoff of permutation test, were also investigated in exactly the same way. For the maximum size of TADs, both the $R^2$ and Jaccard index show a rising pattern while the parameter increases from 0.5 to 1.5 Mb, but as this parameter grows above 1.5 Mb, these two metrics no longer have a significant improvement (Fig. 5e, h). Thus, it is not recommended to tune the maximum size of TADs too large, since it cannot obviously improve the goodness of model fitting and the reproducibility across replicates, but consumes extra computing resources. As for the $p$-value cutoff, the reproducibility across replicates was also examined while the cutoff changes from 0.03 to 0.07 near the default value 0.05 (Fig. 5i). It is shown that the median value of Jaccard index is always larger than 0.84, and do not fluctuate too much in different cases, telling the robustness of called TADs to this parameter.

To evaluate the computational performance of TADfit in running time, we compared the execution time of TADfit with the other five TAD callers on replicates for chromosome 8 of GM12878 cell line at resolutions of 50K, 25K, and 10K (Supplementary Fig. 22). It is shown that the running time of all the methods increases with the growth of the size of contact matrix fed into them, the efficiency of our TADfit is close to that of the other methods except for OnTAD and SpectralTAD, and slightly better than TADtree and TADpole.

In summary, TADfit is a promising computational tool for the structural analysis of genome at TAD level. Beyond the existing methods, it has an ability to handle Hi-C contact matrix replicates and find the underlying hierarchical TADs with more comprehensive structures across them, such as partially overlapping ones. The analysis results support our hypothesis that each IF in contact matrix reflects the cumulative effect of hierarchical TADs in which it is fallen. It is expected that TADfit helps to get a deeper insight into the genome structure at TAD level, especially with continuous accumulation of replicate Hi-C data.

## Methods

**Datasets**. Both simulated and experimental Hi-C data are involved to compare the performance of TADfit and the other five methods in identification of hierarchical TADs. For simulated data, a modified version of the strategy given by Forcato[27] and Lun[49] was used to generate a total of 50 Hi-C contact matrices, where partially overlapping TADs are allowed. These simulated contact matrices can be divided into five groups according to noise level. Each level has ten samples, five of which are contact matrix replicates with partially overlapping TADs, and the other five are replicates without partially overlapping TADs (Supplementary Table 3). For experimental data, the Hi-C contact matrices covering different experimental designs (e.g., different restriction enzymes), cell lines (GM12878, IMR90, and K562), and resolutions (50K, 25K, and 10K) were derived from three Hi-C studies[3,4,50]. To prepare these contact matrices, 15 experimental Hi-C samples were downloaded from Juicer data archive at https://bcm.app.box.com/v/aidenlab/ in the form of a highly compressed binary file .hic (Supplementary Table 4). Then the contact matrices were extracted from these files using the Dump command provided by a java-based program called juicer_tools[51], and were normalized by means of iterative correction and eigenvector decomposition (ICE)[52] as well as log counts per million (logCPM). Beyond Hi-C data, some histone marks (H3K36me3, H3K4me3, H3K27ac, and H3K27me3), architectural proteins (CTCF, Smc3, and Rad21), regulatory elements (promoter and enhancer) and RNA-seq data associated with chromatin activity were taken into account to validate the biological relevance of identified TADs. They can be downloaded from ENCODE[53], UCSC genome browser[54] (Supplementary Table 5), or captured online through New WashU Epigenome Browser[55].

**Preparation of candidate hierarchical TADs**. The candidate hierarchical TADs on Hi-C contact matrix replicates are prepared using a strategy of arbitrary assembly of TAD boundaries. Considering that contact matrix replicates share the same set of TAD boundaries, a pseudo contact matrix is generated by calculating the geometric mean per matrix element across replicates, and the boundaries on diagonal of the pseudo contact matrix are called using TopDom (V. 0.0.2) with the settings recommended in its manual. Then, the boundaries are assembled in all-possible pairs, and the chromatin domain between each pair of boundaries is regarded as a candidate TAD. In this way, the candidate TADs can cover all the possible hierarchical structures at TAD level, including disjoint, nested and partially overlapping ones. In addition, the prepared candidate hierarchical TADs can be optionally screened according to the size threshold given by user.

**Modeling the relationship between IFs and candidate hierarchical TADs**. Assume that each IF in contact matrix reflects the cumulative effect of hierarchical TADs in which it is fallen. Based on this, a linear regression model is proposed to describe the relationship between IFs and candidate TADs. Given a $m \times m$ contact matrix with a total of $n$ candidate TADs, the proposed model has the following form:

$$\mathbf{y} = \mathbf{Xb} + \boldsymbol{\varepsilon} \tag{1}$$

where $\mathbf{y} \in \mathbb{R}^{\frac{m(m+1)}{2}}$ is an artificial vector derived from IFs in the lower triangle and diagonal of normalized contact matrix on a log scale. $\mathbf{X} \in \mathbb{R}^{\frac{m(m+1)}{2} \times n}$ is a designed matrix in which $X_{ij} = \log \frac{m}{d+1}$ if the $i$th IF in $\mathbf{y}$ falls into the $j$th candidate TAD, otherwise $X_{ij} = 0$, and $d$ indicates the bin distance of the $i$th IF to the diagonal. $\mathbf{b} \in \mathbb{R}^n$, $b_j \geq 0$ is a vector consisting of unknown coefficients for the candidate TADs. $\boldsymbol{\varepsilon} \in \mathbb{R}^{\frac{m(m+1)}{2}}$ denotes an error vector in which $\varepsilon_i \in N(0, \sigma^2)$ is assumed to be independently and identically distributed. Thus, the unknown coefficients for the candidate TADs can be estimated by solving the following optimization problem:

$$\underset{\mathbf{b}}{\arg\min} \quad \left\{ \|\mathbf{y} - \mathbf{Xb}\|_2^2 + l\|\mathbf{b}\|_1 \right\} \tag{2}$$
$$s.t. \qquad b_j \geq 0$$

where the first term is a quadratic loss function, and the second term indicates $L_1$ regularization which helps to sparse the objective coefficients.

The model can be extended to profile the hierarchical TADs from multiple contact matrix replicates (Supplementary Note 1). Given a context with $k$ replicates, an artificial matrix $\mathbf{Y} = (\mathbf{y}_1, \mathbf{y}_2, ...\mathbf{y}_k)$ is composed of $k$ artificial vectors in Eq. (1), and the coefficients for candidate TADs in these replicates are denoted as $\mathbf{B} = (\mathbf{b}_1, \mathbf{b}_2, ...\mathbf{b}_k)$. Considering that the hierarchical TAD structures across replicates are consensus, there is $\mathbf{b}_1 = \mathbf{b}_2 = ... = \mathbf{b}_k$, and consequently the unknown coefficients for these hierarchical TADs can be estimated in the following form:

$$\underset{\mathbf{B}}{\arg\min} \quad \left\{ \sum_{j=1}^{k} (\|\mathbf{Y}_{.j} - \mathbf{XB}_{.j}\|_2^2 + l\|\mathbf{B}_{.j}\|_1) \right\} \tag{3}$$
$$s.t. \qquad \mathbf{b}_1 = \mathbf{b}_2 = ... = \mathbf{b}_k$$
$$B_{ij} \geq 0$$

where the subscript $.j$ corresponds to the $j$th column of a matrix.

**Solving the model by FTRL**. The optimization problem above is solved with the help of an online machine learning algorithm called FTRL. This algorithm can be seen as a hybrid of Forward-Backward Splitting (FOBOS)[56] and Regularized Dual Average (RDA)[57], which centers stabilizing regularization in the manner of FOBOS but handles $L_1$ regularization in the manner of RDA, so that a better tradeoff between accuracy and sparsity can be achieved[38]. The online learning solver FTRL is chosen in this study due to two main reasons. One is online. Both the artificial matrix $\mathbf{Y}$ and designed matrix $\mathbf{X}$ are fed into FTRL row by row, so that much less memory is needed. The other is sparsity. Only non-zero elements in $\mathbf{X}$ are needed to be engaged in updating the coefficients of the hierarchical TADs, which speeds up the online learning process. Compared with classical convex optimization methods, these two advantages make our solver much more computationally economical, since $\mathbf{Y}$, especially designed matrix $\mathbf{X}$, usually have ultra-high dimensions and considerable degree of sparsity, due to the consideration of multiple contact matrix replicates and all-possible hierarchical TADs. For example, we have $\mathbf{X} \in \mathbb{R}^{49,715,406 \times 233,586}$ while dealing with five contact matrix replicates for chromosome 1 of GM12878 at 25K resolution. It is unacceptable for most servers to perform a calculation of such a large matrix with limited memory, which may be much worse at higher resolutions. Fortunately, the online learning solver FTRL allows to be fed row by row from $\mathbf{X}$, so that only the memory that can handle $\mathbf{X}' \in \mathbb{R}^{1 \times 233,586}$ is enough. Besides, only about 0.20% elements of the designed matrix $\mathbf{X}$ in this example are non-zero, since each IF in $\mathbf{Y}$ falls into a limited number of related hierarchical TADs in all-possible ones. That helps to give full play to the advantage of FTRL in speeding up the online learning process relying on the sparsity of $\mathbf{X}$. During the iterative optimization using FTRL, there are three hyperparameters that need to be tuned, including $\alpha$ and $\beta$ combined for per-coordinate learning rate of online gradient descent, and $l$ for the strength of $L_1$

regularization. For the first two hyperparameters, according to the original proposer of FTRL, $\alpha$ can vary depending on the datasets and features, and $\beta$ is usually good enough when set to one[39]. For the last hyperparameter, $l$ controls the sparsity of weights by penalizing the absolute value of them. Thus, in this study, $\beta$ is fixed to one, and the other two hyperparameters $\alpha$ and $l$ can be tuned near default values to obtain an approximate optimal solution. To examine how significantly the tunning of $\alpha$ and $l$ affect iterative optimization, the robustness of TADfit to the two hyperparameters is discussed in the "Discussion" section. To determine the significant hierarchical TADs, the regression coefficients are screened with a threshold to exclude overly impossible ones, then a right-tailed permutation test is used to assess how significantly the remaining coefficients are larger than zero with a null hypothesis that the coefficients are equal to zero. This permutation test is implemented by means of a *permTS* function from R package *perm* (1.0.0.0)[58], and the hierarchical TADs with *p*-value <0.05 are determined as the final significant hierarchical TADs.

**Simulation of contact matrix**. Simulated Hi-C contact matrices were generated using a modified version of the procedures proposed by Forcato[27] and Lun[49]. We did not change the basic assumptions and settings used to generate contact matrix in the original method. Interaction frequencies are generated by randomly sampling from a negative binomial distribution with a dispersion parameter and a mean value $\mu$. The dispersion parameter is always set to 0.01, and the mean value $\mu_{ij}$ in the $i$th row and $j$th column of contact matrix is defined as the sum of three signal components:

$$\mu_{ij} = \begin{cases} K_t(i-j+p)^c & \text{if bin pair } (i,j) \text{ is inside a TAD} \\ 0 & \text{otherwise} \end{cases}$$
$$+ \begin{cases} K_d(i-j+p)^c & \text{if bin pair } (i,j) \text{ is part of chromatin loop} \\ 0 & \text{otherwise} \end{cases} \quad (4)$$
$$+ \begin{cases} K_{\text{Noise}} & \text{if bin pair } (i,j) \text{ is sampled as a noise} \\ 0 & \text{otherwise} \end{cases}$$

where the first signal component is added to reflect the effects on the interaction frequency considering whether a bin pair $(i, j)$ is inside a TAD, the second component is added to account for points of chromatin loop ($K_d = 2K_t = 56$), and the third component is considered to account for random noise ($K_{\text{Noise}} = 2$). Besides, the power decay of interaction frequency is considered in the first two components, the decay rate $c$ is set to −0.69, and the prior value $p$ is set to one according to the original procedures. In the light of this way, TADs can be determined by randomly segmenting the chromosomal coordinates, and chromatin loops and noises are decided by randomly sampling all bin pairs at a certain percentage (0.01% for chromatin loops and varying percentages for different noise levels), so that a simulated contact matrix with configured TADs, chromatin loops and noises can be produced. For different contact matrix replicates, the interaction frequencies in them are simulated using the same TADs and chromatin loops but different noises.

Our modifications to the original procedures focus on the generation of partially overlapping TADs. In the original simulation, the hierarchical TADs are prepared by stacking three layers of artificial TADs. The first layer of TADs are disjoint, and are produced with a size ranging from 3 to 20 bins, they are also termed base TADs. Then the second and third layers are generated by randomly removing 25% of the TAD boundaries on the lower layer, so that some meta-TADs on higher layer appears by merging the sub-TADs on the lower layer. In this way, only nested hierarchical TADs can be generated without considering partially overlapping structures. To generate hierarchical TADs with partially overlapping structures, herein we modified the original procedure for preparation of the second layer leaving the procedures for preparation of the first and third layers unchanged. In detail, after 25% of TAD boundaries on the first layer are randomly removed, a certain percentage (set to 15% by default) of the base TADs are picked out. Then a coupled of meta-TADs are constructed for each picked TAD, one is defined by the lower boundary of the picked TAD and the upper boundary of its upstream base TAD, and the other is defined by the upper boundary of the picked TAD and the lower boundary of its downstream base TAD, so that the two constructed meta-TADs are adjacent and partially overlapped in the region of the picked TAD. While all the three layers are completed, a contact matrix with partially overlapping TADs can be prepared. It is worth noting that we only need to set the percentage to zero, a contact matrix without partially overlapping TADs can be simulated.

In this study, Hi-C contact matrices with and without partially overlapping TADs were both simulated with default configurations. For contact matrices with partially overlapping TADs, five levels of random noises, including 4%, 8%, 12%, 16%, and 20%, were considered, and each level has five replicates, so that a total of 25 contact matrices with partially overlapping TADs were simulated. For contact matrices without partially overlapping TADs, there were also a total of 25 contact matrices simulated with the same noise level and the same number of replicates.

**Generation of contact matrix with different sequencing depths**. Based on the existing experimental contact matrices covering different cell lines (GM12878, IMR90, and K562) and resolutions (50K, 25K, and 10K), extra Hi-C contact matrices with different levels of sequencing depths are generated with the help of a down-sampling procedure[40]. Briefly, each experimental contact matrix replicate is converted into a set of pairwise individual interactions without considering the

zero-valued elements, leaving a pairwise interaction vector of length N, where N is the sum of individual elements of the contact matrix. Then a given number of pairwise interactions (1/2*N, 1/4*N, 1/8*N, and 1/16*N) are sampled from this vector by a uniformly sampling procedure. Finally, the chosen interactions are re-binned into a new contact matrix with a fixed sequencing depth. Thus, for each replicate of each cell line at a specific resolution, a total of four contact matrices with the corresponding four different levels of sequencing depths (1/2, 1/4, 1/8, and 1/16) can be obtained.

**Normalization of contact matrix**

*ICE*. ICE is an implicit approach for Hi-C normalization. It attempts to make all bins of contact matrix equally visible using a matrix-balancing strategy[52]. In ICE, the systematic biases between two bins are considered as the product of their individual biases and the maximum likelihood solution for the individual biases is obtained by an iterative correction procedure, yielding a normalized matrix. In this paper, the ICE normalization method was implemented with an R package HiTC (v. 1.24.0)[59] from Bioconductor.

*logCPM transformation*. The contact matrices normalized by ICE are still different in library size from each other, even at the same resolution level. In order to remove the bias caused by sequencing depth and adjust the value of replicates to the same scale, a transformation of log counts per million (logCPM) was performed here. The transformation is given by:

$$O_{ij} = \log 10 \left( \frac{I_{ij}}{L} \times 10^6 + s \right) \quad (5)$$

where $I_{ij}$ and $O_{ij}$ are the interaction frequency in the $i$th row and $j$th column of contact matrix before and after transformation, respectively. $L$ denotes the library size estimated by the sum of the lower triangular matrix, and $s$ was set to one to ensure that $O_{ij}$ is non-negative.

**Evaluation of the accuracy of TAD callers**

*Jaccard index*. Jaccard index is used to score the similarity between two sets of TADs. Given a set of ground-truth TADs $\mathbf{T} = (T_1, T_2, T_3, \dots)$ and a set of called TADs $\mathbf{C} = (C_1, C_2, C_3, \dots)$, the Jaccard index between TAD $T_i$ in $\mathbf{T}$ and TAD $C_j$ in $\mathbf{C}$ is defined as the number of the intersection bins over the number of the union bins (IoU) of the two TADs:

$$\text{Jaccard}(T_i, C_j) = \frac{|T_i \cap C_j|}{|T_i \cup C_j|}. \quad (6)$$

Based on this, the Jaccard index of TAD $C_j$ to TAD set $\mathbf{T}$ is given as follows:

$$\text{Jaccard}(\mathbf{T}, C_j) = \max_i \text{Jaccard}(T_i, C_j). \quad (7)$$

In the same way, the Jaccard index of TAD $T_i$ to TAD set $\mathbf{C}$ can also be obtained. And finally, the Jaccard index between two sets of TADs $\mathbf{T}$ and $\mathbf{C}$ is defined as the average value of the two groups of Jaccard indexes above.

*Modified Jaccard index*. Modified Jaccard index is proposed to estimate how well the TADs observed at a lower resolution are reproducible at a higher resolution. Given a set of TADs $\mathbf{A} = (A_1, A_2, A_3, \dots)$ called at a lower resolution and a set of TADs $\mathbf{B} = (B_1, B_2, B_3, \dots)$ called at a higher resolution. The Jaccard index of $A_i$ to $\mathbf{B}$ is calculated as Eq. (7), and their average value is defined as the modified Jaccard index of $\mathbf{A}$ to $\mathbf{B}$. Thus, if $\mathbf{A}$ is a subset of $\mathbf{B}$, the value of modified Jaccard index will be one, which means that the TADs called at a lower resolution are fully reproduced at a higher resolution.

*F1 score*. F1 score is also used to quantify the accuracy of different methods for TAD identification. Different from Jaccard index, both the precision and recall are considered in it. Considering the ground-truth TAD set $\mathbf{T}$ and the called TAD set $\mathbf{C}$, the precision $p$ is defined as the ratio of the number of common TADs of $\mathbf{T}$ and $\mathbf{C}$ to the number of TADs in $\mathbf{C}$:

$$p = \frac{|\mathbf{T} \cap \mathbf{C}|}{|\mathbf{C}|} \quad (8)$$

and the recall $r$ is defined as the ratio of the number of common TADs of $\mathbf{T}$ and $\mathbf{C}$ to the number of TADs in $\mathbf{T}$:

$$r = \frac{|\mathbf{T} \cap \mathbf{C}|}{|\mathbf{T}|}. \quad (9)$$

Then the F1 score can be calculated in light of its definition. It is noted that a TAD in $\mathbf{T}$ is considered to be the same one as a TAD in $\mathbf{C}$ if the IoU between them is larger than 90%, since a small deviation of the boundaries of two TADs does not means dissimilarity.

**Competing methods**. A total of 12 other computational methods for TAD identification are involved in this paper, from which five methods, including TADtree, 3DNetMod, OnTAD, SpectralTAD, and TADpole, are selected for comparative analysis (Supplementary Table 6). Among them, TADtree is the first publicly

available method to identify hierarchical TADs to the best of our knowledge, and the other four methods, including 3DNetMod, OnTAD, SpectralTAD, and TAD-pole, are the recent progresses in exploring the hierarchical structure of genome at TAD level. The parameter configurations of our TADfit and these competing methods are given in Supplementary Note 2.

**Reporting summary**. Further information on research design is available in the Nature Research Reporting Summary linked to this article.

## Data availability

The sources for all the Hi-C data, histone marks, architectural proteins, regulatory elements, and RNA-seq data involved in this study can be found in Supplementary Tables 3–5. The source data underlying Figs. 2a–b, 3f–h and 5b–i is provided as Supplementary Data 1. And all datasets are available from the corresponding author on reasonable request.

## Code availability

The source code is available at https://github.com/lhqxinghun/TADfit, including R package TADfit as well as the scripts for simulation, normalization, and down-sampling. These codes can also be found on Zenodo (https://doi.org/10.5281/zenodo.6528680)[60].

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

## Acknowledgements

The authors wish to thank Hongkai Ji's research group at Johns Hopkins Bloomberg School of Public Health and high-performance computing platform of Xi'an Jiaotong University. This work was financially supported by the National Natural Science Foundation of China under Grant 61602367.

## Author contributions

H.L. and Q.P. conceived and designed the method. E.L. and Y.L. implemented the TADfit algorithm. E.L. and T.W. performed data analysis. H.L. and J.H. wrote the paper. H.L. and Q.P. supervised the overall study. All authors read and approved the final manuscript.

## Competing interests

The authors declare no competing interests.
