## [Peer Review File · Communications Biology]

Reviewers' comments:

Reviewer #1 (Remarks to the Author):

Here the authors propose a new method called TADfit to identify hierarchical TADs on Hi-C data. Compared with existing methods, the unique features of this approach are: 1) it is able to make use of information from multiple replicates to improve the detection power; 2) it is able to identify partially overlapping TADs. Overall, I find this is an interesting approach and the identified TAD structures indeed look good based on the data demonstrated in this work. However, I would like the authors to perform additional analysis to show how robust their approach is and why we need to care about those partially overlapping TADs.

Below are my detailed comments:

1. Can these partially overlapping TADs be stably identified in different replicates? How many TADs are partially overlapping each other in human genome? What are the genomic/epigenomic features of the overlapping regions? The authors should also discuss more on the significance of these partially overlapping TADs, for example, what regulatory role can they play during development?
2. The authors extensively evaluated the robustness of the method to two hyperparameters of the FTRL machine learning algorithm. However, two other parameters, the maximum size of TADs and the p-value cutoff of the permutation test, might also affect the identified TADs greatly and need to be evaluated and discussed as well. As a proof, the whole matrix was identified as a TAD in supplementary figure 10, however, this can be definitely avoided by setting the maximum size of TADs.
3. How does the running time of TADfit change with the contact matrix size? In my test, the software ran extremely slow (>12 hours with default parameters) even for the smallest chromosome (chr21) of IMR90 Hi-C at 10kb resolution.

Reviewer #2 (Remarks to the Author):

The author proposed a multivariate linear regression model for profiling hierarchical chromatin domains. The manuscript is well written and compact with results. However, the following points may be noted:

1. The domain is enriched with a dozen of methods such as TADtree, TADDom etc. The author may explain what is the main advantages of their method than the others.
2. Follow-The-Regularized-Leader (FTRL) is utilized to estimate the weights of candidate TADs. The reader may be interested to know the rationale behind the selection of this optimization technique. What are the hyperparameter of FTRL and how these are tuned in the algorithm?
3. How the simulated Hi-C data is created. Author may explain the details of creation of the simulated data.
4. What are the parameters for other methods? For fair comparison author may provide a separate subsection describing the hyperparameters of the competing methods.
5. Author may describe the datasets in a separate subsection.
6. "...significant ones can be screened out by a permutation test"--- how the permutation test is performed? What are the hypothesis?
7. Author describes several literatures in the introduction, however only five methods are selected for comparison. Why the other methods are not considered for comparison purpose?
8. It will be helpful for the reader to see the method in an algorithm format.
9. How the quality of TADs is measured? For biological validation author may consider the method described in the TADtree paper (enrichment of Chip-Seq derived binding sites of several proteins and chromatin marks).

Point-by-point response to reviewers' comments

(Comments are in **black**, responses are in **blue**, figures are inserted for your convenience)

First of all, we would like to thank the reviewers for your time and attention to our work, especially your comments and suggestions to improve our manuscript for publication. With your kindly help, we have revised the manuscript carefully, and highlighted the changes within the revised manuscript and Supplementary Information, which we hope to meet with your approval. The following are our response to the comments.

Reviewers' comments:

Reviewer #1 (Remarks to the Author):

Here the authors propose a new method called TADfit to identify hierarchical TADs on Hi-C data. Compared with existing methods, the unique features of this approach are: 1) it is able to make use of information from multiple replicates to improve the detection power; 2) it is able to identify partially overlapping TADs. Overall, I find this is an interesting approach and the identified TAD structures indeed look good based on the data demonstrated in this work. However, I would like the authors to perform additional analysis to show how robust their approach is and why we need to care about those partially overlapping TADs.

Below are my detailed comments:

1. Can these partially overlapping TADs be stably identified in different replicates? How many TADs are partially overlapping each other in human genome? What are the genomic/epigenomic features of the overlapping regions? The author should also discuss more on the significance of these partially overlapping TADs, for example, what regulatory role can they play during development?

Response: Thanks for your comment. To investigate how stably the partially overlapping TADs can be identified, contact matrix replicates were fed into our TADfit individually, and the Jaccard index of identified partially overlapping TADs between replicates was calculated (Fig. 5b). It is shown that the median value of the index is always higher than 0.60 in all the cases covering different cell lines (GM12878, IMR90, K562) and resolutions (50K, 25K), and the highest value can reach up to 0.87. The number of called partially overlapping TADs in human genome may change, especially under different resolutions and different parameter tunings. Herein we tried to scan the whole genome of GM12878 at 25K resolution using our TADfit under default parameter setting in the way of multi-replicate input, the numbers of both partially overlapping TADs and the other TADs for every chromosome are given (Supplementary Fig. 19). It is found that about 10.95% of the called TADs are partially overlapping ones. The results tell that the partially overlapping structure at TAD level may be considerable in spatial organization of human genome. This partially overlapping structure was taken into account in this study based on the following two considerations. One is that the super-resolution microscopy image of chromatin with nanometer-scale precision shows that TAD structures in a bulk Hi-C contact matrix can be considered as the ensemble average of TAD-like domains in thousands of single cells¹, and the boundaries of these TAD-like domains in single cells present a high degree of cell-to-cell variations¹⁻³, whereas it is suggested that the hierarchical TADs in bulk Hi-C contact matrix are unlikely to be perfectly nested, but also partially overlapped. The

Fig. 5b. Jaccard index of partially overlapping TADs across replicates. The contact matrix replicates for chromosome 1 of three cell lines (GM12878, IMR90 and K562) at resolutions of 50K and 25K were fed into TADfit individually.

Supplementary Fig. 19. Numbers of partially overlapping TADs and the other TADs called by TADfit on a genome-wide scale. The hierarchical TADs were called on contact matrix replicates (GSM1551550_HIC001 - GSM1551554_HIC005) for all chromosomes of GM12878 at resolution of 25K.

other one is that the partially overlapping structure between TADs has been considered in a few existing computational methods or their future improvement plans. 3DNetMod⁴ is a graph-theory-based approach which declares the ability to handle partially overlapping TADs. OnTAD⁵ does not have this ability, but the method takes partially overlapping structure into account and tries to approximate it with nested TADs, for example, the partially overlap between two adjacent TADs can be roughly recaptured with three nested TADs. And SuperTAD⁶, which is devoted to the identification of nested TADs, regards the detection of partially overlapping TADs as a challenging future work.

For the genomic/epigenomic features of partially overlapping regions, the histone marks (H3K36me3, H3K4me3, H3K27ac and H3K27me3) and active genes (FPKM > 5) were taken into account (Supplementary Fig. 20). Generally, a difference in the enrichment of these features can be seen between partially overlapping regions and the other regions by examining all the results throughout different cell lines (GM12878 and K562), that is, the enrichment of active histone marks (H3K36me3, H3K4me3 and H3K27ac) and active genes in partially overlapping regions trend to be higher than that of the other regions, while the repressive histone mark (H3K27me3) does the opposite. That can be explained, since active epigenetic states and highly expressed genes are reported to be more enriched in inner TADs than in outer TADs⁷⁻¹⁰, and the partially overlapping regions are usually inner

Supplementary Fig. 20 Comparison of biological significance between partially overlapping regions and the other regions of hierarchical TADs called by TADfit. The hierarchical TADs were called **a** on contact matrix replicates (GSM1551550_HIC001 -GSM1551554_HIC005) for chromosome 1 of GM12878, and **b** on contact matrix replicates (GSM1551619_HIC070 - GSM1551623_HIC074) for chromosome 1 of K562 at 25K resolution, respectively. The .bigwig files of ChIP-seq for histone marks (H3K36me3, H3K4me3, H3K27ac and H3K27me3) and the .tsv files for RNA-seq data were downloaded from ENCODE (Supplementary Table 3). The average ChIP-seq signals as well as density of active genes (FPKM > 5) within both partially overlapping regions and the other regions of hierarchical TADs were shown.

Supplementary Fig. 21 Comparison of regulatory elements between partially overlapping regions and the other regions of hierarchical TADs called by TADfit. The hierarchical TADs were called **a** on contact matrix replicates (GSM1551550_HIC001 -GSM1551554_HIC005) for chromosome 1 of GM12878, and **b** on contact matrix replicates (GSM1551619_HIC070 - GSM1551623_HIC074) for chromosome 1 of K562 at 25K resolution, respectively. The .bed files for genomic annotations were downloaded from UCSC genome browser (Supplementary Table 3). The density of four categories of regulatory elements (active promoter, weak promoter, strong enhancer and weak enhancer) within both partially overlapping regions and the other regions of hierarchical TADs was shown.

parts in a hierarchy of TADs. Furthermore, although the mechanisms underlying hierarchical TADs, especially partially overlapping ones, in gene expression regulation remain unclear^{11,12}, the promoters and enhancers, especially active promoters and strong enhancers, are found to be favored within partially overlapping regions (Supplementary Fig. 21). That suggests the high-intensity

interaction between enhancers and promoters within these regions and support the enrichment difference in active genes mentioned above.

Action(s) taken: According to your comment, the partially overlapping TADs have been further discussed in terms of the reproducibility across replicates, number in the whole human genome, genomic/epigenomic features and potential regulatory role. The statements were appended to the end of the first paragraph of “Discussion” section, and the figures mentioned above were also added into the manuscript (Fig. 5b) and Supplementary Information (Supplementary Figs. 19 - 21) accordingly.

2. The authors extensively evaluated the robustness of the method to two hyperparameters of the FTRL machine learning algorithm. However, two other parameters, the maximum size of TADs and the p-value cutoff of the permutation test, might also affect the identified TADs greatly and need to be evaluated and discussed as well. As a proof, the whole matrix was identified as a TAD in supplementary figure 10, however, this can be definitely avoided by setting the maximum size of TADs.

Response: Thanks for your reminder. The two parameters, including the maximum size of TADs and p-value cutoff of permutation test, were taken into account and discussed in the same way. For the maximum size of TADs, the robustness was examined while it changes from 0.5 Mb to 2.5 Mb, considering that the median size of TADs in mammals is reported to be about hundreds of kilobases^{13,14}. Generally, both the R^2 and Jaccard index show a rising pattern while the parameter increases from 0.5 Mb to 1.5 Mb, but as this parameter grows above 1.5 Mb, these two metrics no longer have a significant improvement (Fig. 5e, h). Thus, it is not recommended to tune the maximum size of TADs too large, since it cannot obviously improve the goodness of model fitting and the reproducibility across replicates, but consumes extra computing resources. As for the p-value cutoff, the reproducibility across replicates was also evaluated while it changes from 0.03 to 0.07 near the default value 0.05 (Fig. 5i). It is shown that the median value of Jaccard index is always larger than 0.84, and do not fluctuate too much in different cases, telling the robustness of called TADs to this parameter. In addition, the special case in Supplementary Fig. 10 that the whole matrix was identified as a TAD by TADfit can be indeed avoided by tuning down the maximum size of TADs. But in fact, this TAD does not occupy the whole simulated matrix, just about half of the matrix, since only partial simulated matrix was drawn in this figure. That is also the case in Fig. 2 and Supplementary Figs. 1 - 9. Obviously, this misunderstanding is attributed to the unclear legends below these figures, which are misleading our readers.

Fig. 5e. Curve of R^2 (mean \pm SD) with different maximum sizes of TADs. **h** Jaccard index of called TADs across replicates with different maximum sizes of TADs. **i** Jaccard index of called TADs across replicates with different p-value cutoffs. The contact matrix replicates for chromosome 1 of three cell lines (GM12878, IMR90 and K562) at 25K resolution were separately fed into TADfit with each cell line as a group.

Action(s) taken: According to your comment, the three subgraphs (Fig. 5e, h, i) were integrated into Figure 5 of the manuscript to show how significantly the tuning of these two parameters, including the maximum size of TADs and p-value cutoff of permutation test, affect model fitting and reproducibility across replicates, and the corresponding statements were also appended to the end of the last paragraph of “Discussion” section to discuss the robustness of TADfit to these two parameters. Besides, all the legends of Fig. 2 and Supplementary Figs. 1 - 10 were modified to specify the location of visualized part in original simulated contact matrix.

3. How does the running time of TADfit change with the contact matrix size? In my test, the software ran extremely slow (>12 hours with default parameters) even for the smallest chromosome (chr21) of IMR90 Hi-C at 10kb resolution.

Response: Thanks for your reminder. Our TADfit does take such a long time when dealing with the whole chromosome, the unexpected inefficiency is mainly due to the use of parameter *maxscalesize* with a default value of 0.5. For example, when handling a chromosomal segment with a length of 5 Mb, the corresponding maximum size of TADs becomes $5\text{Mb} \times 0.5 = 2.5\text{Mb}$, which may be reasonable for mammals in most cases. However, for the whole chromosome, even the smallest chromosome 21 of IMR90 at 10K resolution, the corresponding maximum TAD size can reach up to $48\text{Mb} \times 0.5 = 24\text{Mb}$ (about 2400 bins), which is too large and consumes extra computing resources. Considering that the maximum and minimum sizes of TADs are separately set to about 200 bins and 3 bins in many competing methods, such as TADtree¹⁵, OnTAD and SpectralTAD⁸ (Supplementary Methods), we tried to adopt the same configurations by adjusting the two parameters *maxscalesize* and *minscalesize* to $4.17e-2$ and $6.25e-4$ respectively, and reran TADfit on the whole chromosome 21 of IMR90 at 10K resolution using a Ubuntu 14.04 LTS with Intel(R) Xeon(R) E5-2609 @ 1.70 GHz CPU. Unlike before, only about 112 minutes are needed now.

Responsive Fig. 1 Comparison of running time between TADfit and the other five hierarchical TADs callers, including TADtree, 3DNetMod, OnTAD, SpectralTAD and TADpole, on the same computing platform. The comparison was conducted on contact matrix replicates (GSM1551599_HIC050 - GSM1551604_HIC055) for the whole chromosome 8 of IMR90 at resolutions of 50K, 25K and 10K, and the corresponding dimensions of input contact matrices are 2920×2920 , 5840×5840 and 14600×14600 , respectively. The contact matrix replicates at the same resolution level were all fed into TADfit at one time, but into the other five callers individually, since only TADfit is a multi-replicate method. The maximum and minimum sizes of TADs in TADfit were separately set to 200 bins and 3 bins by default, which is in line with the configurations of most of the other callers (Supplementary Methods).

Action(s) taken: According to your kindly reminder, the two parameters *maxscalesize* and *minscalesize* were replaced by *maxsize* and *minsize*, which are no longer the ratios to the entire matrix, but the absolute sizes of TADs with default values of 200 bins and 3 bins, respectively. These changes have been declared in a new subsection titled “Configurations of TADfit and competing methods” of Supplementary Information, and updated in a new version of TADfit which is available on Github. In addition, the running time of TADfit under the new default configurations was also compared with that of the other five hierarchical TADs callers involved in the manuscript, including TADtree, 3DNetMod, OnTAD, SpectralTAD and TADpole¹⁶ (Responsive Fig. 1). It is shown that the running time of all the methods increases with the growth of the size of contact matrix fed into them, the efficiency of our TADfit is close to that of the other methods except for OnTAD and SpectralTAD, and even slightly better than TADtree and TADpole.

Reviewer #2 (Remarks to the Author):

The author proposed a multivariate linear regression model for profiling hierarchical chromatin domains. The manuscript is well written and compact with results. However, the following points may be noted:

1. The domain is enriched with a dozen of methods such as TADtree, TopDom etc. The author may explain what is the main advantages of their method than the others.

Response: Thanks for your comment. A total of twelve other methods for TAD identification, including Directionality Index, HiCseg¹⁷, Insulation Score¹⁸, TopDom¹⁹, ClusterTAD²⁰, TADtree, GMAP²¹, CaTCH²², 3DNetMod, OnTAD, SpectralTAD and TADpole, are involved in the manuscript. The first five are designed for detection of TAD boundaries, while the latter seven take into account the hierarchical structures of TADs.

Action(s) taken: According to your comment, the respective characteristics of these twelve methods are outlined in the second paragraph of “Introduction” section. Besides, compared with the existing methods, the main advantages of our model TADfit were summarized into two issues and described in the last paragraph of “Introduction” section, and a new “Conclusion” section was also added into the manuscript to declare the advantages of TADfit according to the results and discussion.

2. Follow-The-Regularized-Leader (FTRL) is utilized to estimate the weights of candidate TADs. The reader may be interested to know the rationale behind the selection of this optimization technique. What are the hyperparameter of FTRL and how these are tuned in the algorithm?

Response: Thanks for your reminder. TADfit is a multivariate linear regression model, the weights of candidate hierarchical TADs can be estimated by solving a minimum objective function consists of a quadratic loss and L_1 regularization. To get the solution of this optimization problem, we had tried several traditional solvers, such as Lasso, but all failed, and finally chose the Google’s gradient-based online learning solver FTRL due to two main reasons. One is online. The artificial matrix \mathbf{Y} , especially designed matrix \mathbf{X} , usually have ultra-high dimensions, since multiple contact matrix replicates and all-possible hierarchical TADs are involved in this model. For example, we have $\mathbf{X} \in \mathbb{R}^{49,715,406 \times 233,586}$ while dealing with five contact matrix replicates for chromosome 1 of GM12878 at 25K resolution. It is unacceptable for most servers to perform a calculation of such a large matrix with limited memory, which may be much worse at higher resolutions. Fortunately, in the online learning solver FTRL, \mathbf{X} are fed row by row, so that much less memory is needed, that is, only the memory that can handle $\mathbf{X}' \in \mathbb{R}^{1 \times 233,586}$ is enough. The other is sparsity. Only non-zero elements in \mathbf{X} are needed to be engaged in updating the coefficients of the hierarchical TADs, which speeds up the learning process. In the above example, only about 0.20% elements of the designed matrix \mathbf{X} are non-zero, since each IF in \mathbf{Y} falls into a limited number of related hierarchical TADs in all-possible ones. That helps to give full play to the advantage of FTRL in speeding up the online learning process relying on the sparsity of \mathbf{X} . These two reasons make it realistic to solve the model with limited computing resources, which prompts us to choose the online learning solver FTRL. There are three hyperparameters that need to be tuned for FTRL, including α and β combined for per-coordinate learning rate of online gradient descent, and l for the strength of L_1 regularization. For the first two hyperparameters, according to the original proposer of FTRL, α can vary depending on the datasets and features, and β is usually good enough when set to one. For the last

hyperparameter, l controls the sparsity of weights by penalizing the absolute value of them. Thus, in this study, β is fixed to one, α and l are separately set to 0.01 and 2 by default, and can be tuned near the default values to obtain an approximate optimal solution.

Action(s) taken: According to your comment, we explained the two main reasons in more detail, and added the example mentioned above into “Solving the model by FTRL” subsection of the manuscript. It is expected that the rationale behind the selection of FTRL could be clearer for readers. Besides, the three hyperparameters α , β and l as well as how they should be tuned were further described in the middle of “Solving the model by FTRL” subsection. And the discussion on the robustness to the two hyperparameters α and l has also been modified in the last paragraph of “Discussion” section of the manuscript.

3. How the simulated Hi-C data is created. Author may explain the details of creation of the simulated data.

Response: Thanks for your comment. The simulated Hi-C data was created by a modified version of the Hi-C data simulation procedures given by Forcato, M. et al ²³ and Lun, A. T. et al ²⁴. Our modifications focus on the generation of partially overlapping TADs in Hi-C contact matrix, which is not considered in the original procedure.

Action(s) taken: According to your comment, we modified the “Simulation of contact matrix” subsection of Supplementary Information, the details of Hi-C data simulation was described to make the modified procedure clearer. Besides, the R code for data simulation as well as simulated Hi-C data are freely available at <https://github.com/anonymous-doubleblind/TADfit/tree/main/data/simdata>.

4. What are the parameters for other methods? For fair comparison author may provide a separate subsection describing the hyperparameters of the competing methods.

Response: Thanks for your reminder. The five competing methods, including TADtree, 3DNetMod, OnTAD, SpectralTAD and TADpole, have their own different hyperparameters. It is really true as you pointed out that a separate subsection describing the hyperparameters of these competing methods should be given.

Action(s) taken: According to your comment, a new subsection titled “Configurations of TADfit and competing methods” was added into the Supplementary Information. In this subsection, the hyperparameters of the five competing methods and our TADfit are all described in detail. Besides, the URLs for the executable codes of these methods are also given.

5. Author may describe the datasets in a separate subsection.

Response: Thanks for your comment. A separate subsection for datasets can really make the structure of the manuscript clearer for readers.

Action(s) taken: According to your comment, the descriptions about datasets have been reorganized into a new separate subsection titled “Datasets” and added to the beginning of “Materials and methods” part of the manuscript. In this subsection, simulated and experimental Hi-C data as well as histone marks, architectural proteins, regulatory elements and RNA-seq data are all described. Besides, the sources of all the above datasets are listed in “Supplementary Datasets” section of Supplementary Information.

6. “...significant ones can be screened out by a permutation test”--- how the permutation test is performed? What are the hypothesis?

Response: Thanks for your comment. A permutation test was used to assess how significantly the regression coefficients are greater than zero. Thus, the null hypothesis is that the coefficients are equal to zero, and the alternative hypothesis is that the coefficients are greater than zero. In the implementation, the right-tailed permutation test was performed with the help of the *permTS* function from R package *perm* (1.0.0.0)²⁵.

Action(s) taken: According to your comment, the type of permutation test was declared throughout the manuscript, that is, right-tailed permutation test. Besides, the more detailed descriptions of this right-tailed permutation test were added into the end of “Solving the model by FTRL” subsection of the manuscript.

7. Author describes several literatures in the introduction, however only five methods are selected for comparison. Why the other methods are not considered for comparison purpose?

Response: Thanks for your comment. As mentioned above, a total of twelve other methods for TAD identification are involved in “Introduction” section. The first five are designed for detection of TAD boundaries, while the latter seven go a step further to take into account the hierarchical structures of TADs. In this study, the methods are selected for comparison based on the following two considerations. On the one hand, the first five methods (Directionality Index, HiCSeq, Insulation Score, TopDom and ClusterTAD) can only be used to identify TAD boundaries, while our TADfit mainly focuses on the profile of complex hierarchical structures of TADs. In this sense, it may be unnecessary to show the superiority of our method by comparing with these early methods. On the other hand, from the latter seven methods (TADtree, GMAP, CaTCH, 3DNetMod, OnTAD, SpectralTAD and TADpole) that have the ability to identify hierarchical TADs, there are five methods, including TADtree, 3DNetMod, OnTAD, SpectralTAD and TADpole, selected for comparison, since they are considered to be representative and competitive. In detail, TADtree is the first publicly available method to identify hierarchical TADs (cited up to 136 times), and the other four methods, including 3DNetMod, OnTAD, SpectralTAD and TADpole, are the latest progress in analyzing the hierarchical structure of TADs (Responsive Table 1). It is expected that the five selected methods help to illustrate the advance of our method in this specific field.

Responsive Table 1. Summary of the twelve methods for TAD identification involved in “Introduction” section. The five methods selected for comparison are indicated in bold.

Method	Type	Journal	Year
Directionality Index	TAD boundary	Nature	2012
HiCSeq	TAD boundary	Bioinformatics	2014
Insulation Score	TAD boundary	Nature	2015
TopDom	TAD boundary	Nucleic Acids Research	2016
ClusterTAD	TAD boundary	BMC Bioinformatics	2017
TADtree	Hierarchical TADs	Bioinformatics	2016
GMAP	Hierarchical TADs	Nature Communications	2017
CaTCH	Hierarchical TADs	Genome Research	2017
3DNetMod	Hierarchical TADs	Nature Methods	2018
OnTAD	Hierarchical TADs	Genome Biology	2019
SpectralTAD	Hierarchical TADs	BMC Bioinformatics	2020
TADpole	Hierarchical TADs	Nucleic Acids Research	2020

8. It will be helpful for the reader to see the method in an algorithm format.

Response: Thanks for your comment. An algorithm format can indeed make our method much clearer for readers.

Action(s) taken: According to your comment, a new subsection titled “Algorithm of TADfit model and FTRL solver” has been added into Supplementary Information to describe our proposed method in an algorithm format. In this subsection, the pseudo code for TADfit model and FTRL solver is given in Supplementary Algorithm 1. Besides, the statements in “Materials and methods” section of the manuscript have also been modified accordingly.

9. How the quality of TADs is measured? For biological validation author may consider the method described in the TADtree paper (enrichment of Chip-Seq derived binding sites of several proteins and chromatin marks).

Response: Thanks for your comment. The enrichment of ChIP-seq derived binding sites of architectural proteins (CTCF, Smc3 and Rad21) and chromatin mark (H3K4me3) were taken into account to investigate the quality of the hierarchy of called TADs. In the implementation, just like the method described in the TADtree paper, the number of ChIP-seq peaks within 50 Kb of TAD boundaries was calculated. Then, considering that the TAD boundaries along the diagonal of contact matrix are determined by TopDom, and our model is devoted to the anatomy of hierarchical structures based on these boundaries, herein we take one more step forward to conduct a comparative analysis of biological significance of TAD boundaries between different hierarchical levels across three cell lines (GM12878, IMR90 and K562) on a genome-wide scale (Supplementary Fig. 17). It is found that the enrichment of these ChIP bindings at TAD boundaries is significantly enhanced as the hierarchical level of boundaries increases (right-tailed paired *t*-test *p*-value < 5.04e-4 for level 2 versus level 1, right-tailed paired *t*-test *p*-value < 6.22e-5 for level ≥ 3 versus level 2), which is in line with the observations of other studies^{7,8,10} where the TAD boundaries with higher levels are believed to be more biologically significant. These tell the quality of the hierarchy of TADs called by TADfit

Supplementary Fig. 17 Enrichment of architectural proteins and histone mark within 50 Kb of multiple-level boundaries of hierarchical TADs called by TADfit on a genome-wide scale. The hierarchical TADs were called **a** on contact matrix replicates (GSM1551550_HIC001 -GSM1551554_HIC005) for all chromosomes of GM12878, **b** on contact matrix replicates (GSM1551599_HIC050 - GSM1551604_HIC055) for all chromosomes of IMR90, and **c** on contact matrix replicates (GSM1551619_HIC070 - GSM1551623_HIC074) for all chromosomes of K562 at 25K resolution, respectively. The level of a TAD boundary is defined using the terminology introduced by An, L. et al.⁵, that is, a TAD boundary belonging to a single TAD is regarded as a first-level boundary, and the second-level and third-level boundaries correspond to the boundaries that are shared by two and three hierarchical TADs, respectively. The peak files of ChIP-seq for architectural proteins (CTCF, Smc3 and Rad21) and histone mark (H3K4me3) were downloaded from ENCODE (Supplementary Table 3), and the number (mean \pm SD) of ChIP-seq peaks within 50 Kb of TAD boundaries at three different levels was calculated.

to some extent. It is worth noting that the biological relevance of the regression coefficients for hierarchical TADs given by TADfit is examined in terms of histone marks, gene expression level and A/B compartment in “Biological significance of TADs profiled by TADfit” subsection of the manuscript.

Action(s) taken: According to your comment, the number of ChIP-seq peaks at multi-level TAD boundaries across three cell lines (GM12878, IMR90 and K562) on a genome-wide scale was visualized and added into Supplementary Information (Supplementary Fig. 17). The corresponding analysis of these results has also been appended to the end of “Biological significance of TADs profiled by TADfit” subsection of the manuscript.

Other revisions:

1. The abstract was refined to 147 words, no more than the required maximum word count of 150.
2. All the floating-point values throughout the revised files were limited to two decimal places.
3. The sentence at the end of the last paragraph of “Introduction” section was modified due to the consideration of architectural proteins and regulatory elements, as well as the analysis of TAD boundaries at different levels.
4. The compositions of the manuscript were renumbered due to the addition of new content, such as “Conclusion” part and “Datasets” subsection.
5. Standard deviation is added to fully describe the statistics together with the original mean.
6. The original “Solving with FTRL” subsection was removed from Supplementary Information due to the partial repetition with the new subsection “Algorithm of TADfit model and FTRL solver”.
7. A few grammatical errors in the text were corrected.
8. A new version of TADfit was updated on Github with URL unchanged.

References:

- 1 Bintu, B. *et al.* Super-resolution chromatin tracing reveals domains and cooperative interactions in single cells. *Science* **362**, eaau1783 (2018).
- 2 Nagano, T. *et al.* Single-cell Hi-C reveals cell-to-cell variability in chromosome structure. *Nature* **502**, 59-64 (2013).
- 3 Stevens, T. J. *et al.* 3D structures of individual mammalian genomes studied by single-cell Hi-C. *Nature* **544**, 59-64 (2017).
- 4 Norton, H. K. *et al.* Detecting hierarchical genome folding with network modularity. *Nat. Methods* **15**, 119-122, doi:10.1038/nmeth.4560 (2018).
- 5 An, L. *et al.* OnTAD: hierarchical domain structure reveals the divergence of activity among TADs and boundaries. *Genome Biol.* **20**, 282, doi:10.1186/s13059-019-1893-y (2019).
- 6 Zhang, Y. W., Wang, M. B. & Li, S. C. SuperTAD: robust detection of hierarchical topologically associated domains with optimized structural information. *Genome Biol.* **22**, 1-20 (2021).
- 7 Wang, X., Cui, W. & Peng, C. HiTAD: detecting the structural and functional hierarchies of topologically associating domains from chromatin interactions. *Nucleic Acids Res.* **45**, e163 (2017).
- 8 Cresswell, K. G., Stansfield, J. C. & Dozmorov, M. G. SpectralTAD: an R package for defining a hierarchy of Topologically Associated Domains using spectral clustering. *BMC Bioinf.* **21**, 1-19 (2020).

- 9 Luzhin, A. V. *et al.* Quantitative differences in TAD border strength underly the TAD hierarchy in *Drosophila* chromosomes. *Journal of Cellular Biochemistry* **120**, 4494-4503 (2019).
- 10 Du, G. *et al.* The hierarchical folding dynamics of topologically associating domains are closely related to transcriptional abnormalities in cancers. *Comput. Struct. Biotechnol. J.* **19**, 1684-1693 (2021).
- 11 Sikorska, N. & Sexton, T. Defining functionally relevant spatial chromatin domains: it is a TAD complicated. *J. Mol. Biol.* **432**, 653-664 (2020).
- 12 Franke, M. *et al.* Formation of new chromatin domains determines pathogenicity of genomic duplications. *Nature* **538**, 265-269 (2016).
- 13 Dixon, J. R. *et al.* Topological domains in mammalian genomes identified by analysis of chromatin interactions. *Nature* **485**, 376-380, doi:10.1038/nature11082 (2012).
- 14 Nora, E. P. *et al.* Spatial partitioning of the regulatory landscape of the X-inactivation centre. *Nature* **485**, 381-385, doi:10.1038/nature11049 (2012).
- 15 Weinreb, C. & Raphael, B. J. Identification of hierarchical chromatin domains. *Bioinformatics* **32**, 1601-1609, doi:10.1093/bioinformatics/btv485 (2016).
- 16 Solervila, P., Cusco, P., Farabella, I., Stefano, M. D. & Martirenom, M. A. Hierarchical chromatin organization detected by TADpole. *Nucleic Acids Res.* **48**, e39 (2020).
- 17 Lévy-Leduc, C., Delattre, M., Mary-Huard, T. & Robin, S. Two-dimensional segmentation for analyzing Hi-C data. *Bioinformatics* **30**, i386-i392, doi:10.1093/bioinformatics/btu443 (2014).
- 18 Crane, E. *et al.* Condensin-driven remodelling of X chromosome topology during dosage compensation. *Nature* **523**, 240-244 (2015).
- 19 Shin, H. *et al.* TopDom: an efficient and deterministic method for identifying topological domains in genomes. *Nucleic Acids Res.* **44**, e70, doi:10.1093/nar/gkv1505 (2016).
- 20 Oluwadare, O. & Cheng, J. ClusterTAD: an unsupervised machine learning approach to detecting topologically associated domains of chromosomes from Hi-C data. *BMC Bioinf.* **18**, 480-480 (2017).
- 21 Yu, W., He, B. & Tan, K. Identifying topologically associating domains and subdomains by Gaussian Mixture model And Proportion test. *Nat. Commun.* **8**, 535, doi:10.1038/s41467-017-00478-8 (2017).
- 22 Zhan, Y. *et al.* Reciprocal insulation analysis of Hi-C data shows that TADs represent a functionally but not structurally privileged scale in the hierarchical folding of chromosomes. *Genome Res.* **27**, 479-490 (2017).
- 23 Forcato, M. *et al.* Comparison of computational methods for Hi-C data analysis. *Nat. Methods* **14**, 679-685, doi:10.1038/nmeth.4325 (2017).
- 24 Lun, A. T. & Smyth, G. K. diffHic: a Bioconductor package to detect differential genomic interactions in Hi-C data. *BMC Bioinf.* **16**, 258, doi:10.1186/s12859-015-0683-0 (2015).
- 25 Fay, M. P. & Shaw, P. A. Exact and asymptotic weighted logrank tests for interval censored data: the interval R package. *Journal of statistical software* **36** (2010).

REVIEWERS' COMMENTS:

Reviewer #1 (Remarks to the Author):

I think the authors did a great job addressing my previous comments. I do not have any further concerns.

Reviewer #2 (Remarks to the Author):

The authors have satisfactorily addressed all issues indicated earlier. The manuscript is much improved now. Therefore, the manuscript may be accepted now after fixing a few typos.

Point-by-point response to reviewers' comments

(Comments are in **black**, responses are in **blue**)

We would like to thank the reviewers for your time and attention to our work, especially your valuable comments and suggestions. With your kindly help, our manuscript has been improved substantially.

Reviewers' comments:

Reviewer #1 (Remarks to the Author):

I think the authors did a great job addressing my previous comments. I do not have any further concerns.

Response: Thank you very much for your insightful comments and generous approval. We appreciate your kindly help to improve our work.

Reviewer #2 (Remarks to the Author):

The authors have satisfactorily addressed all issues indicated earlier. The manuscript is much improved now. Therefore, the manuscript may be accepted now after fixing a few typos.

Response: Thank you so much for your insightful comments, which have greatly helped to improve our work. Besides, the manuscript has been carefully examined, and all the typos found have been fixed.